# Genetic Evaluation and Population Structure of Jiangsu Native Pigs in China Revealed by SINE Insertion Polymorphisms

**DOI:** 10.3390/ani12111345

**Published:** 2022-05-25

**Authors:** Xiaoyan Wang, Enrico D’Alessandro, Chenglin Chi, Ali Shoaib Moawad, Wencheng Zong, Cai Chen, Chengyi Song

**Affiliations:** 1College of Animal Science & Technology, Yangzhou University, Yangzhou 225009, China; wxyan@yzu.edu.cn (X.W.); mz120191039@yzu.edu.cn (C.C.); ali.shoeib@agr.kfs.edu.eg (A.S.M.); zongzone@126.com (W.Z.); chencai9596@hotmail.com (C.C.); 2Unit of Animal Production, Department of Veterinary Science, University of Messina, 98168 Messina, Italy; edalessandro@unime.it; 3Department of Animal Production, Faculty of Agriculture, Kafrelsheikh University, Kafrelsheikh 33516, Egypt

**Keywords:** retrotransposon insertion polymorphism, SINE, Jiangsu native pig, genetic evaluation

## Abstract

**Simple Summary:**

In a previous study, 30 SINE-RIPs were applied for population genetic analysis in 7 Chinese miniature pig populations and approved effectively in the genetic distances and breed-relationships between these populations. There are abundant indigenous pigs famous across the world for their prolificacy in the Jiangsu Province of eastern China, such as Meishan, Erhualian. Since pork production relies on limited commercial breeds such as Landrace, Large White, and Duroc pigs, characterized by maximizing productivity in intensive production systems, these indigenous pigs are nowadays decrease sharply. The genetic characterizations of these resources are essential requirements for the development of conservation, selection, and sustainable utilizations. Therefore, SINE-RIPs were selected to evaluate the genetic variation and population structure of Jiangsu pig populations and the results may assist with the conservation and utilization of these native pig populations.

**Abstract:**

Short interspersed nuclear elements (SINEs), one type of retrotransposon, are considered to be ideal molecular markers due to their wide distribution in the genome, high copy number, and high polymorphism. Preliminary studies have identified more than 35,000 SINE-retrotransposon insertion polymorphisms (RIPs) in the pig genome. In this study, 18 SINE-RIPs were used to evaluate the genetic variation and population structure of seven native pig populations and two crossbreeds in the Jiangsu Province of China. Two commercial pig breeds (Duroc and Large White) and one Italian native breed (Sicilian Black pig) were selected as the control. The results showed that all 18 SINE-RIPs were polymorphic among these pigs. The Jiangsu native pig populations (Erhualian, Fengjing, Middle Meishan, Mi, Shawutou, Small Meishan, and Huai) were shown to be more polymorphic than the crossbreeds (Sushan and Sujiang) and external breeds (Sicilian Black pig, Large White, and Duroc) based on the expected heterozygosity and polymorphic information content values. Some native pigs, including Small Meishan, Mi, Middle Meishan, and Erhualian, had a higher degree of inbreeding according to the *F*_IS_ values. Based on the neighbor-joining tree, all of the Jiangsu native pig populations formed one branch, while the three external pig breeds formed the other branches, with the two crossbreeds containing more than 50% external pig ancestry. The Huai pigs were independent of the other Jiangsu native pigs but shared a common ancestor with Sujiang and Mi. The results provide a new perspective on the population structure of these native pig breeds and will assist with the conservation and utilization of Chinese native pigs.

## 1. Introduction

There are abundant indigenous pigs in the Jiangsu Province of eastern China and they are famous across the world for their prolificacy, high-quality pork, and moderate temper [1]. The most prolific breed, Meishan, has contributed to the improvement of many commercial pig breeds [2]. Meishan pigs can be divided into two subpopulations, Small Meishan and Middle Meishan, according to their body size. Other breeds around Taihu Lake include Erhualian, Mi, Fengjing, and Shawutou. Huai pigs, an ancient breed that has existed for 2000 years in the northern Jiangsu Province, have been identified as having an important influence on the evolution of other pigs in Jiangsu Province [3].

Due to their high prolificacy, high-quality pork, and moderate temper, researchers have focused on the genetic evaluation, population structure, and conservation of some of the breeds around Taihu Lake. Chang et al., (1999) used the randomly amplified polymorphic DNA (RAPD) technique with 13 decamer primers and found that pigs around Taihu Lake exhibited a relatively low level of genetic variation and poor genetics [4]. Fan et al., (2002) used microsatellites and found that the genetic variability of the indigenous pigs descended in the following order: Mi, Erhualian, Shawutou, Middle Meishan, and Small Meishan [5]. Also, Wang (2015) investigated the genetic diversity of six breeds around Taihu Lake and found that the genetic diversity of the Meishan breed was the greatest, whereas that of Fengjing was the lowest. The neighbor-joining (NJ) tree revealed that the Middle Meishan and Small Meishan pigs were clustered into two separate clades, whereas the Fengjing and Jiaxing Black pigs were grouped into a single clade using a genome reducing and sequencing (GGRS) approach [1]. DNA next-generation sequencing confirmed that the single nucleotide polymorphisms (SNPs) of the Erhualian and Mi and Fengjing and Jiaxing Black breeds were closely clustered in a principal component analysis (PCA) [6]. Additionally, the number of effective alleles per locus (Ne) of the Small Meishan pig breed was smaller than those of the other Chinese pig breeds, indicating that the Small Meishan pig underwent stronger selection pressure than the other breeds [7]. Furthermore, the analysis using next-generation sequencing provided a basis for setting conservation priorities for these seven indigenous pig breeds, with a focus on the total genetic diversity and allelic diversity [8]. Up to now, few studies have been reported that use retrotransposon insertion polymorphisms (RIPs) for genetic evaluation and to study the population structure of these Jiangsu pig breeds.

Retrotransposons are one class of transposable elements (TE) that can move freely in the genome via “copy-and-paste” mechanisms and induce allelic diversity [9,10]. One-half to one-third mammalian genome is derived from transposable elements and retrotransposons are the most common types of TE [11]. Therefore, retrotransposons lead to rich genetic diversity in many different species and contribute to genome evolution. Accordingly, several molecular markers based on the polymorphism of retrotransposon insertion patterns were developed, including SSAP (sequence-specific amplification polymorphism), IRAP (inter-retrotransposon amplified polymorphism), REMAP (retrotransposon-microsatellite amplified polymorphism), and RIP markers [12,13]. As one type of these molecular markers, RIP markers have been used in several species including humans [14] and dogs [15], among others. One type of retrotransposon, short interspersed nuclear elements (SINEs), is dispersed throughout eukaryotic genomes and can be used for diagnosing common ancestry among host taxa due to their enormous volume and the irreversible, independent nature of their insertion [16]. SINEs range in size from 150 to 500 bp and are widely dispersed among the chromosomes including gene-rich regions [17,18]. They lack the machinery that is necessary for self-mobilization and use the enzymatic machinery of autonomous retrotransposons (LINEs [long interspersed nuclear elements]) [17,19]. The RIPs which are based on SINE, also named SINE-RIPs, are suitable for determining genetic diversity and variety differentiation in plants [20,21,22]. Additionally, SINE insertion polymorphisms have been applied extensively for human population genetic analysis [23].

Preliminary studies have determined that SINEs occupy 11.05% of the pig genome [24]. Using RepeatMasker, three young SINE subfamilies (SINEA1, SINEA2, and SINEA3) were screened among sixteen assembled pig genomes downloaded from the NCBI whole genome sequencing database. All putative SINE-RIPs were verified by local BLAST and PCR and more than 35,000 SINE-RIPs have been identified [25]. Sixteen SINE-RIPs from 16 chromosomes were applied to construct phylogeny tree in 23 pig breeds and the results were generally consistent with the geographical distributions of native pig breeds in China [25]. Futhermore, the genetic distances and breed-relationships between seven Chinese miniature pigs using 30 SINE-RIPs were also agree with their evolutions and geographic distributions [26]. In this study, 18 involved SINE-RIPs that were identified in the preliminary studies were used to genetically evaluate and construct the population structure of seven native pig populations (Mi, Erhualian, Fengjing, Shawutou, Middle Meishan, Small Meishan, and Huai) and two crossbreeds (Sujiang and Sushan) in Jiangsu Province.

## 2. Materials and Methods

### 2.1. Sample Collection

Ear or blood samples were collected and used as biological material for DNA extraction. The samples included seven native populations (Mi, Erhualian, Fengjing, Shawutou, Middle Meishan, Small Meishan, and Huai) and two crossbreeds (Sujiang and Sushan) in the Jiangsu Province of China. These pigs were kept in indigenous conservation breeding farms. Two commercial breeds (Large White and Duroc) and one European native breed (Sicilian black pig) [27] were used as the control in this study. The general information for these stocks, including the origin and sample size are presented in Appendix A. The photos of the twelve pig populations and their geographical distributions are shown in Figure 1.

### 2.2. DNA Extraction and SINE-RIPs Genotyping

Genomic DNA was extracted from the ear or blood samples of each pig using the TIANamp Genomic DNA Extraction Kit (TIANGEN Biotech Co. Ltd., Beijing, China). The concentration and quality of the DNA were determined using ultraviolet spectrophotometry (Implen GmbH, Munchen, Germany). For amplification, 18 SINE-RIP markers were selected from a previous study [25] according to their degree of polymorphism. The primers were synthesized by Beijing Liuhe Huada Gene Technology Co. Ltd., and the sequence information is summarized in Appendix A.

The DNA samples were amplified using the Amp9600 Thermal Cycler (ABI, New York, NY, USA). The polymerase chain reaction (PCR) mixture contained 40 ng genomic DNA, 10 ul 2 × Taq Master Mix buffer (Vazyme, Nanjing, China), and 10 pmol of each primer. The PCRs were performed using the following conditions: an initial denaturation step at 94 °C for 5 min, followed by 30 cycles of denaturation at 94 °C for the 30 s, annealing at 55–60 °C for 30 s, and extension at 72 °C for 1 min. A final extension of 10 min at 72 °C was also included. The PCR products were loaded onto a 1.0% agarose gel in 1 × TAE buffer and visualized using a UV fluorescence system (Tanon, Shanghai, China).

### 2.3. Statistical Analyses

The POPGEN1.32 computer package was used to calculate the allele frequencies, number of alleles per locus (Na), Ne, observed heterozygosity (*H*_o_), expected heterozygosity (*H*_e_), fixation indices (*F*_IS_, *F*_ST_), and Hardy-Weinberg equilibrium test [28]. The polymorphic information content (PIC) was calculated as PIC = 1 − $\sum xi^{2}-\overset{n-1}{\underset{i=1}{\sum }}\overset{n}{\underset{j=i+1}{\sum }}2pi^{2}pj^{2}$, where *n* is the number of alleles, *pi* is the frequency of the *i*th allele in the population, and *pj* is the frequency of the *j*th allele in the population [29]. The NJ clustering analyses were based on Nei’s genetic distance and the phylogenetic trees were constructed using Mega (version 7.0). The population structure of the twelve pig populations was constructed using the Bayesian method in STRUCTURE (version 2.3.4). The principal component analysis (PCA) was performed using the R statistics package (v. 3.6.3). The delta K values were calculated and the appropriate K value was estimated using the STRUCTURE Harvester program (http://taylor0.biology.ucla.edu/struct_harvest/, accessed on 4 March 2021; [30]). Then, CLUMPP and Distruct were used to account for repeat sampling and to visualize the data, respectively [31,32].

## 3. Results

### 3.1. Evaluation of the SINE-RIPs in Jiangsu Pig Populations

All 18 loci used in this study were polymorphic. Considering that the crossbreed Sujiang consists of 62.5% Duroc and 50% of the Sushan breed is Large White, the Duroc and Large White commercial pig breeds were designated as the control. Additionally, Sicilian Black pigs from Italy were used as a control as an external native pig breed. All of the loci had two alleles and the clear amplified bands are shown in Figure 2. Three genotypes were identified in the electrophoretogram including one long band for a homozygote with a SINE insertion, one short band for a homozygote without a SINE insertion, and two bands for a heterozygote with a long band and a short band. Compared to the commercial pigs, the Jiangsu native pigs and crossbreeds showed more polymorphism in the 18 loci. More than 16 loci showed polymorphism in all populations except for Huai which had only 11 polymorphic loci. Duroc had only one polymorphic locus and Large White had seven polymorphic loci. Sicilian Black had 11 polymorphic loci. The insertion frequency, polymorphism, Hardy-Weinberg equilibrium, *F*_IS_, and *F*_ST_ are listed in Table 1.

### 3.2. Genetic Variability

The *H*_e_ of the 12 populations ranged from 0.0172 to 0.3606, the *H*_o_ ranged from 0.0208 to 0.3854, and the average values were 0.2183 and 0.2203, respectively (Table 2). The average Ne was 1.3640 and it ranged from 1.0243 to 1.6265; whereas the average PIC was 0.1832 and it ranged from 0.0143 to 0.2799. The Ne, PIC, *H*_e_, and *H*_o_ of Shawutou were the highest among the 12 populations, which indicates that this native Jiangsu breed shows evidence of high genetic diversity. However, the genetic parameters of the crossbreed Sushan were the lowest among these Jiangsu pig populations, followed by Sujiang, indicating that these two crossbreeds have low genetic diversities, likely as a result of continuous selection in every generation. The *F*_IS_ value which is the inbreeding coefficient of the population ranged from −0.2308 (Duroc) to 0.1108 (Mi). The *F*_IS_ values of Erhualian, Middle Meishan, Mi, Sujiang, and Small Meishan were positive, indicating that these five populations show a degree of inbreeding.

The heatmap of the *F*_ST_ between each pig population is shown in Figure 3. In general, the Jiangsu pig populations were highly differentiated from the two commercial breeds (Large White and Duroc), Italian native breed (Sicilian black), and two crossbreeds (Sujiang and Sushan). Among the seven Jiangsu native pig populations, the *F*_ST_ ranged from 0.0873 to 0.3069. The lowest *F*_ST_ (0.0873) was recorded between Shawutou and Middle Meishan, and the highest *F*_ST_ (0.3069) was recorded between Huai and Small Meishan. Six populations around Taihu Lake (Erhaulian, Fengjing, Middle Meishan, Small Meishan, Mi, and Shawutou) showed relatively low *F*_ST_ values, indicating that they have a similar genetic background. The *F*_ST_ between the three external breeds (the two commercial breeds and Italian native breed) and Jiangsu native pig populations ranged from 0.3250 to 0.5835. The two crossbreeds (Sujiang and Sushan) and external pig breeds had a relatively lower *F*_ST_ (ranging from 0.0142 to 0.1652) in comparison with a higher FST between the crossbreeds and the Jiangsu native pig populations (ranging from 0.2051 to 0.4691). This may be due to the fact that these two crossbreeds consist of more than 50% external pig breed parentage.

### 3.3. Genetic Distance and Population Structure

Based on Nei’s genetic distance (Table 3), between the Jiangsu native pig populations and external breeds was a longer genetic distance than between the native population and Jiangsu crossbreeds. The distance ranged from 0.2758 to 0.6936 between the Jiangsu native breeds and three external pig breeds and 0.1867 to 0.6091 between the Jiangsu native population and crossbreeds. Between the Jiangsu crossbreeds and external breeds was a small genetic distance. Among the seven Jiangsu native populations, the distance was relatively low, and ranged from 0.0955 (Mi and Middle Meishan) to 0.3639 (Middle Meishan and Huai). Huai was more distantly related to the other Jiangsu native pig populations with a distance between them that ranged from 0.2159 (Mi) to 0.3639 (Middle Meishan). The tendency of the genetic identity among the 12 populations was similar to the genetic distance.

As shown in UPGMA tree based on Nei’s distance (Figure 4A), these twelve populations were divided into two main branches. All of the Jiangsu native pig populations formed one branch, while the Jiangsu crossbreeds and three external pig breeds formed the other branches. Among the Jiangsu native pig breeds, six of the pig populations near Taihu Lake grouped together and Huai formed one branch independently. In another branch, Sujiang formed one subbranch, and Sushan and the other external pig breeds formed the other subbranch.

The population structure of the seven Jiangsu native pig populations, two Jiangsu crossbreeds, and three external breeds was quantified using Structure (Figure 4B). When K = 2–4, Erhualian, Fengjing, Small Meishan, Middle Meishan, Mi, and Shawutou around Taihu Lake shared a common ancestry. When K = 4–6, Huai, Sujiang, and Mi shared a common ancestry. Also, Sushan and the three external breeds formed one population. With the increase in the K value, Sushan was closer to SB and LW.

The principal components classified the Jiangsu native pig populations and the other five breeds into two different groups (Figure 5). In addition, Huai was differentiated from the other six Jiangsu native pig populations. These data agreed with the NJ tree and population structure data described above.

## 4. Discussion

In this study, 18 SINE-RIPs were used to evaluate the population structure of the Jiangsu pig populations. All of the 18 SINE-RIPs were identified as polymorphic using PCR. Several studies have focused on the genetic variation and population structure of some of the Jiangsu native breeds using SNPs [1] and microsatellites [5]. The *H*_e_ values were used to evaluate genetic diversity and were recorded as 0.378, 0.375, 0.382, 0.382, and 0.371 using SNPs, and 0.621, 0.603, 0.429, 0.623, and 0.605 using microsatellites for the Erhualian, Middle Meishan, Small Meishan, Mi, and Shawutou breeds, respectively. In this study, the *H*_e_ values were 0.3195, 0.3178, 0.2567, 0.3042, and 0.3606 for these five pig populations. The values that were determined using SINE-RIPs were far lower than those that were determined using microsatellites and just lower than those that were calculated using SNPs. Presumably, this is due to the fact that SINE-RIPs are biallelic markers with long fragment insertion mutations, while microsatellites are multiple markers and SNPs are single nucleotide variations. Previous studies have identified that the Chinese native pig populations showed richer genetic diversity than external pig breeds [5,33], which also was conducted using SINE-RIPs. Compared to most of the other pigs around Taihu Lake, Small Meishan had a lower PIC and *H*_e_ and a higher *F*_IS_ value; which means that this population showed evidence of low genetic variation. In the mid-1980s, a scheme to create an inbreeding line of Small Meishan was developed in order to increase the population size [5]. Therefore, inbreeding could reduce genetic variation to some extent and could explain the values that were observed. It has been suggested that the genetic heterozygosity of Small Meishan could be enhanced by crossing with Middle Meishan [5]. These two populations differ in body size and therefore, ideally, other steps should be taken to increase genetic variation in the process of conservation.

The *H*_e_ and PIC of all of the external pig breeds were lower than those of the Jiangsu native pig breeds. It is possible that the commercial breeds have undergone more intensive selection than the Chinese native breeds which have been conserved. While the herd sizes of the Sicilian Black pigs might be small, Sujiang and Sushan (two crossbreeds between the Jiangsu native breeds and external pig breeds) also underwent intensive selection during their breeding which may explain why their genetic diversity decreased.

The Chinese native pig and European pig originated from different original wild boars around 8000–10,000 years ago [33,34]. In both the NJ tree and PCA, the external pig and Chinese pig clustered separately. Unexpectedly, Sujiang and Susan clustered with the external pig. Sujiang, which was recognized as a new genetic material for pig breeding by the National Committee of Livestock and Poultry Genetic Resource of China in 2013, is a crossbreed with 62.5% Duroc and 37.5% Jiangsu native breeds to improve growth rate and lean percentage of native pigs [35]. With the same objective, Sushan is a crossbreed with 50% Landrace, 25% Large White, and 25% Jiangsu native breeds, which was recognized as a new genetic material for pig breeding by the National Committee of Livestock and Poultry Genetic Resource of China in 2018 [36]. Potentially, these two crossbreeds possess much more external pig ancestry than the Jiangsu native breeds, which would explain the close genetic distance to external pig breeds.

The greatest span from north to south in the Jiangsu Province is about five hundred kilometers. The Huai pig is an indigenous breed in the northern area, while the other populations occur in the south of Jiangsu Province. The NJ trees and PCA plots showed that the two branches split according to geographical distribution and Huai is independent of the other Jiangsu pig populations. According to the history of the formulation of these pig breeds, Erhualian is the offspring of Mi and Dahualina (extinct) in Changzhou [5]. Using SNPs and microsatellites, Erhualian had the closest relationship with the Mi pig than the other populations in Jiangsu [1,5]. However, in this study, Mi clustered with Middle Meishan and had a relatively distant relationship with Erhualian. With the popularity of the commercial breeds, the number of Chinese native breeds, especially boars, decreased sharply. Erhualian is preserved in three independent national conservation farms in Jiangsu and Mi nearly went extinct at one point [1]. To increase the boar lineages, a sow without any kinship to the present pig population was used to cross with a present boar and the F1 male offspring were backcrossed with the other sows in the same lineage of the previous sow around three to four times until the excellent male offspring of the new boar lineage contained over 90% ancestry from the sow lineage. During this period, the genetic variation of this pig population was altered due to the introduction of this new boar lineage [37]. The difference in sampling site and the introduction of this new boar lineage could thus potentially explain why Mi clustered with Middle Meishan, and not Erhualina.

The STRUCTURE analysis revealed that Erhualian, Fengjing, Small Meishan, Middle Meishan, Mi, and Shawutou around Taihu Lake shared a common ancestry (K = 2–3) and Huai, Sujiang, and Mi shared a common ancestry (K = 4–6). The Huai pig, even recorded in the Compendium of Materia Medica, is an ancient native pig breed in the north of Jiangsu Province which is part of the Huaibei Plain. As a result of the war and commercial exchanges, the local people emigrated to the south of Jiangsu province, bringing with the Huai pigs on several occasions. Some arrived at the Jintan County and Mi pigs were gradually formed when Huai pig crossed with local pigs [38]. Furthermore, some moved into the mid-east of the Jiangsu Province and Jiangquhai pigs were formed [3]. Jiangquhai is the parent population of Sujiang. Therefore, Huai, Mi, and Sujiang appear to have a common genetic background.

In this study, 18 SINE-RIPs were selected to evaluate the genetic variation and population structure of Jiangsu pig populations in China. The results confirm that they provide an effective, reliable method for evaluating the genetic variation and distance of Jiangsu native and crossed pig populations. The results also will assist with the conservation and utilization of these native pig populations. Compared with the genetic information provided by microsatellites and SNPs using partial loci or whole genomic sequencing data, the SINE-RIPs could provide a simple and low-cost method to evaluate the population structure of different breeds and population.

## 5. Conclusions

In summary, 18 SINE-RIPs were used for genetic evaluation in 7 Jiangsu native pig populations, two crossbreeds, and three external pig breeds in this study. According to the values of *H*_e_ and *H*_o_, the Jiangsu native pig breed populations showed more genetic variation than the crossbreeds and external pig breeds. The *F*_IS_ values of the Erhualian, Middle Meishan, Mi, and Small Meishan were positive, indicating that these four populations have a degree of inbreeding. In both the NJ tree and PCA, the external pigs and Jiangsu native pigs clustered separately. Erhualian, Fengjing, Small Meishan, Middle Meishan, Mi, and Shawutou around Taihu Lake shared a common ancestry, and Huai, Sujiang, and Mi also shared a common ancestry. Therefore, the 18 SINE-RIPs provide an effective, reliable, and simple method for evaluating the genetic variation and population structure of Jiangsu native and crossed pig populations and the results will assist with the conservation and utilization of this native pig population.

## Figures and Tables

**Figure 1 animals-12-01345-f001:**
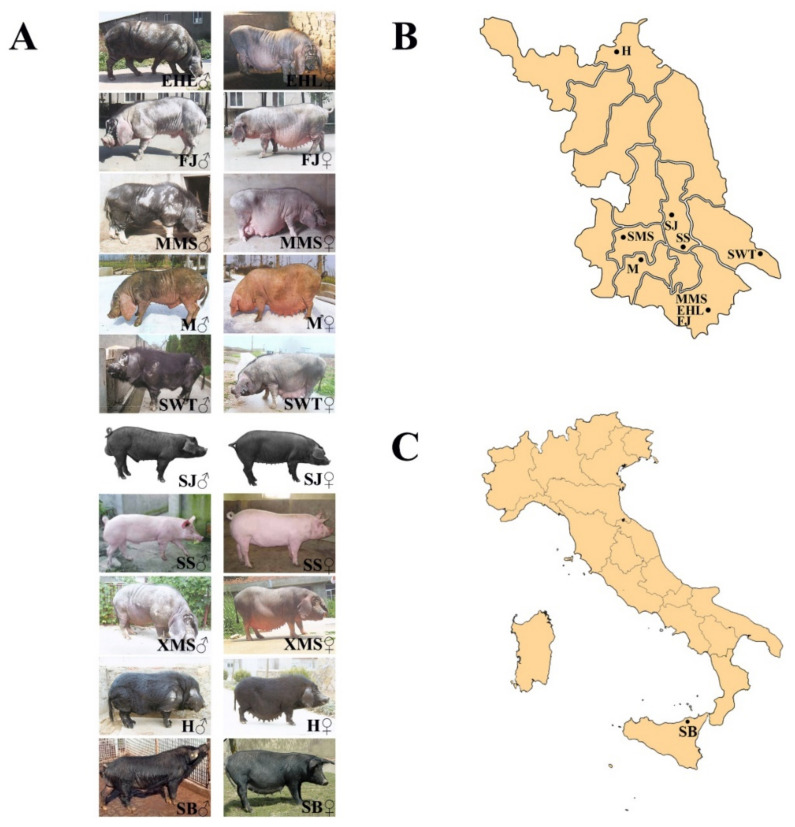
The photos of the ten native pig populations and geographical distribution. (**A**) The photos of the ten native pig populations; (**B**) geographical distribution of nine Jiangsu native pig population in Jiangsu province; (**C**) geographical distribution of Italic native pig breed in Italy. EHL, Erhualian; FJ, Fengjing; MMS, Middle Meishan; M, Mi; SWT, Shawutou; SJ, Sujiang; SS, Sushan; SMS, Small meishan; H, Huai; SB, Sicilian black pig; LW, Large White; DRC, Duroc.

**Figure 2 animals-12-01345-f002:**
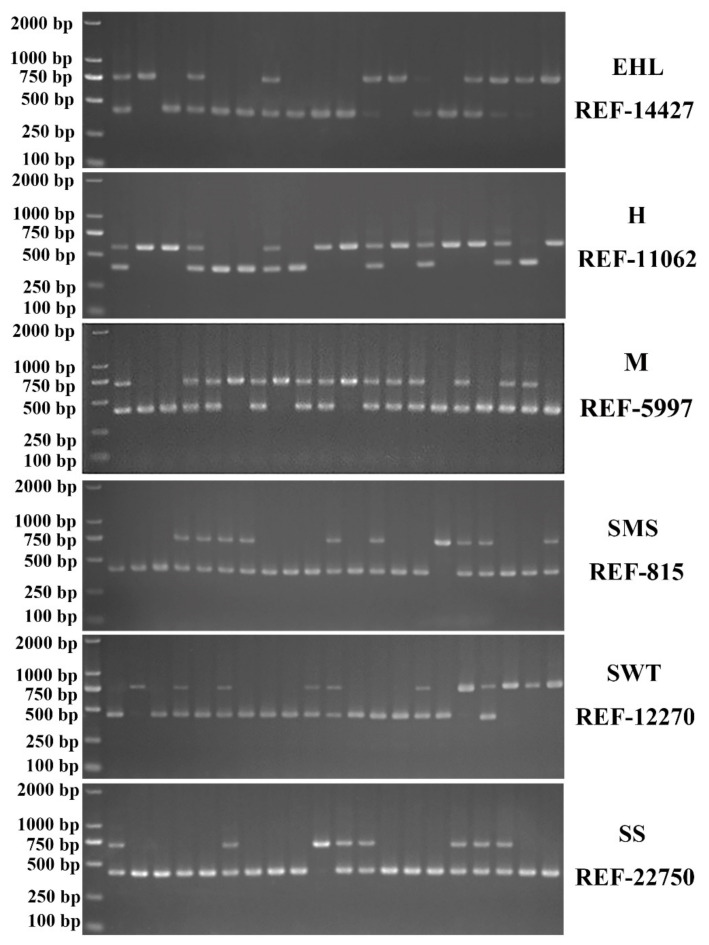
Representative electropherograms of SINE-RIPs detected in some pig breeds. EHL, Erhualian; H, Huai; M, Mi; SMS, Small meishan; SWT, Shawutou; SS, Sushan.

**Figure 3 animals-12-01345-f003:**
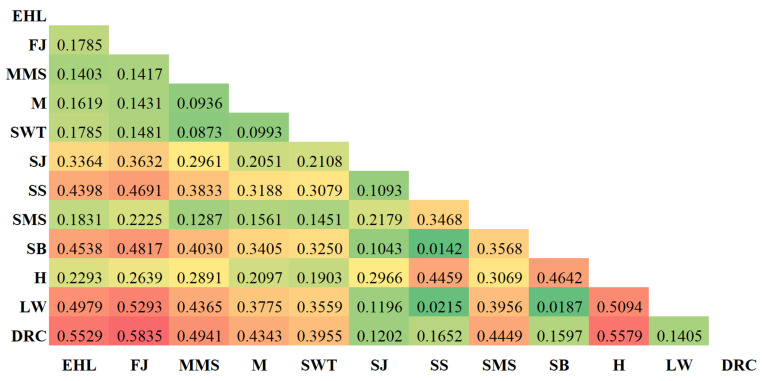
Heatmap of the fixation indices (*F*st) between 12 pig population including seven Jiangsu pig population. The color from red to green indicted that the FST value was decreased. EHL, Erhualian; FJ, Fengjing; MMS, Middle Meishan; M, Mi; SWT, Shawutou; SJ, Sujiang; SS, Sushan; SMS, Small meishan; H, Huai; SB, Sicilian black pig; LW, Large White; DRC, Duroc.

**Figure 4 animals-12-01345-f004:**
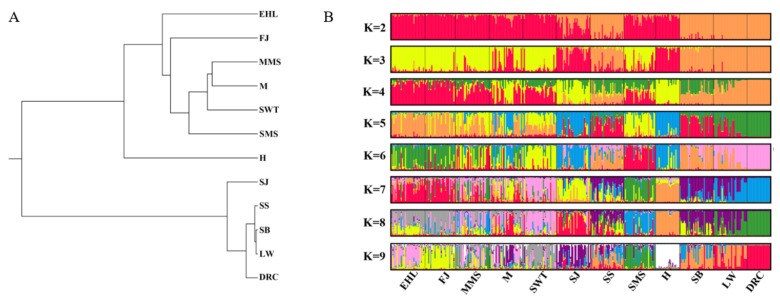
UPGMA tree on Nei’s genetic distances and graphical representation with K 2–9. (**A**) UPGMA tree constructing by Mega7; (**B**) graphical representation with K 2–9 by STRUCTURE software. Different color showed different origination. EHL, Erhualian; FJ, Fengjing; MMS, Middle Meishan; M, Mi; SWT, Shawutou; SJ, Sujiang; SS, Sushan; SMS, Small meishan; H, Huai; SB, Sicilian black pig; LW, Large White; DRC, Duroc.

**Figure 5 animals-12-01345-f005:**
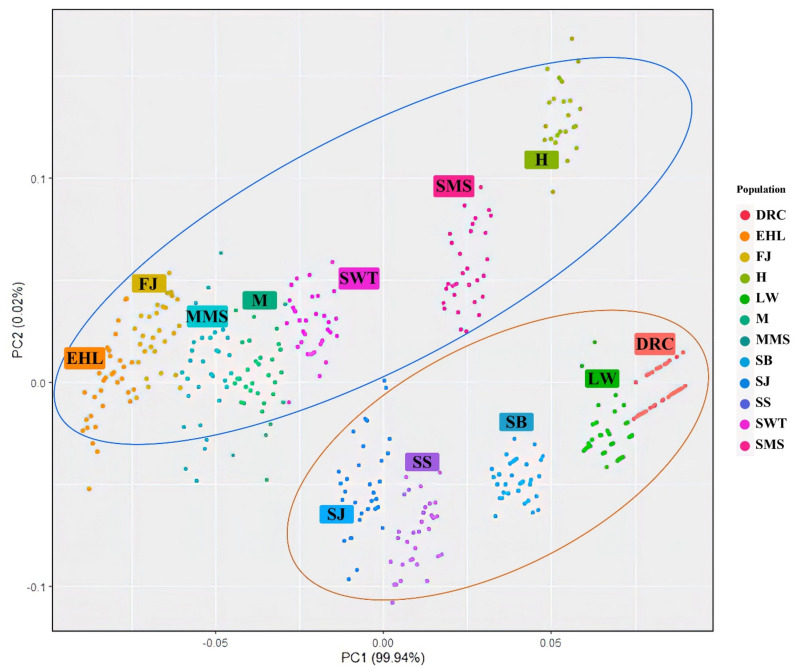
PCA plots for 12 pig populations. Note: EHL, Erhualian; FJ, Fengjing; MMS, Middle Meishan; M, Mi; SWT, Shawutou; SJ, Sujiang; SS, Sushan; SMS, Small meishan; H, Huai; SB, Sicilian black pig; LW, Large White; DRC, Duroc.

**Table 1 animals-12-01345-t001:** Polymorphism of SINE RIPs in 12 pig populations.

SINE-RIPs	Insertion Frequency												No. of Populations Show Polymorphic	No. of Populations Show Hardy–Weinberg Disequilibrium	*F* _IS_	*F* _ST_
	EHL	FJ	MMS	M	SWT	SJ	SS	SMS	H	SB	LW	DRC				
REF-815	1.00	0.02	0.47	0.25	0.19	0.00	0.00	0.22	0.00	0.00	0.00	0.00	5	0	−0.0645	0.559
REF-12270	0.48	0.36	0.53	0.45	0.23	0.02	0.64	0.10	0.00	0.52	0.45	0.00	10	1	0.0599	0.2325
REF-13182	1.00	1.00	0.92	1.00	0.81	0.39	0.17	0.98	1.00	0.05	0.08	0.19	8	0	−0.0894	0.6808
REF-14427	0.58	0.41	0.11	0.16	0.00	0.03	0.06	0.02	0.96	0.02	0.00	0.00	9	2	0.3258	0.5339
REF-16131	0.39	0.57	0.50	0.50	0.72	0.17	0.00	0.13	0.22	0.00	0.00	0.00	8	3	−0.1121	0.3126
REF-16684	0.08	0.40	0.08	0.05	0.06	0.00	0.00	0.30	0.00	0.00	0.00	0.00	6	0	−0.1354	0.2129
REF-18327	0.39	0.62	0.86	0.25	0.36	0.00	0.00	0.45	0.00	0.00	0.00	0.00	6	3	0.0704	0.4313
REF-19717	0.05	0.14	0.33	0.13	0.56	0.03	0.00	0.32	0.13	0.00	0.00	0.00	8	0	−0.0369	0.2385
REF-21609	0.84	0.85	0.94	0.70	0.56	0.27	0.03	0.55	0.67	0.00	0.03	0.00	10	1	−0.0914	0.4959
REF-2929	0.98	1.00	0.98	0.83	0.61	0.25	0.16	1.00	0.37	0.25	0.16	0.00	9	0	−0.1704	0.5683
REF-3719	0.59	0.10	0.25	0.13	0.48	0.08	0.03	0.03	1.00	0.00	0.00	0.00	8	1	−0.0223	0.5148
REF-4531	0.52	0.67	0.14	0.05	0.14	0.06	0.11	0.18	0.00	0.19	0.03	0.00	10	1	−0.1918	0.2789
REF-5597	0.92	0.71	0.59	0.11	0.72	0.14	0.08	0.93	0.44	0.06	0.03	0.00	11	1	−0.0636	0.5081
REF-7445	0.66	0.38	0.48	0.48	0.33	0.30	0.03	0.35	0.44	0.06	0.00	0.00	10	1	−0.1221	0.2137
REF-8430	0.23	0.93	0.41	0.23	0.38	0.00	0.02	0.02	0.39	0.02	0.00	0.00	9	2	−0.0053	0.4219
REF-9435	0.84	0.86	0.47	0.83	0.28	0.31	0.05	0.58	0.61	0.03	0.00	0.00	10	2	0.1531	0.4416
REF-10096	0.53	0.71	0.00	0.41	0.42	0.33	0.16	0.03	0.89	0.20	0.08	0.00	10	1	−0.0309	0.358
REF-11062	0.45	0.97	0.86	0.86	0.98	0.08	0.09	0.23	0.63	0.02	0.00	0.00	10	2	0.0896	0.6186
No. of loci show non-polymorphic	2	2	1	1	1	4	5	1	7	7	11	17	N	N	N	N

Note: EHL, Erhualian; FJ, Fengjing; MMS, Middle Meishan; M, Mi; SWT, Shawutou; SJ, Sujiang; SS, Sushan; SMS, Small meishan; H, Huai; SB, Sicilian black pig; LW, Large White; DRC, Duroc.

**Table 2 animals-12-01345-t002:** Genetic parameters generated by the 18 SINE RIPs.

Breed	Sample Size	*H* _e_	*H* _o_	Polymorphic Information Content (PIC)	Effective Number of Allele (Ne)	*F* _is_
EHL	32	0.3195 ± 0.1999	0.3108 ± 0.2347	0.2467 ± 0.1404	1.5687 ± 0.4102	0.0070 ± 0.2802
FJ	29	0.2984 ± 0.1894	0.3142 ± 0.2409	0.2339 ± 0.1347	1.5069 ± 0.3738	−0.0261 ± 0.2812
MMS	32	0.3178 ± 0.1831	0.2969 ± 0.1661	0.2485 ± 0.1254	1.5515 ± 0.3936	0.0177 ± 0.1464
M	32	0.3042 ± 0.1535	0.2674 ± 0.1832	0.2438 ± 0.1064	1.4922 ± 0.3251	0.1108 ± 0.2999
SWT	32	0.3606 ± 0.1625	0.3854 ± 0.1896	0.2799 ± 0.1147	1.6265 ± 0.3317	−0.0821 ± 0.1746
SJ	32	0.2039 ± 0.1828	0.2240 ± 0.2123	0.1653 ± 0.1360	1.3197 ± 0.3256	0.0744 ± 0.1656
SS	32	0.1242 ± 0.1314	0.1181 ± 0.1147	0.1069 ± 0.1022	1.1688 ± 0.2169	−0.0142 ± 0.1425
SMS	30	0.2567 ± 0.1904	0.2704 ± 0.2351	0.2040 ± 0.1367	1.4238 ± 0.3691	0.0662 ± 0.3972
SB	32	0.1117 ± 0.1582	0.1215 ± 0.1790	0.0925 ± 0.1196	1.1688 ± 0.2783	−0.0718 ± 0. 0988
H	23	0.2357 ± 0.2243	0.2464 ± 0.2400	0.1813 ± 0.1628	1.4123 ± 0.4214	−0.0708 ± 0.1428
LW	32	0.0694 ± 0.1312	0.0677 ± 0.1190	0.0581 ± 0.0991	1.1039 ± 0.2387	−0.0387 ± 0.1027
DRC	32	0.0172 ± 0.0730	0.0208 ± 0.0884	0.0143 ± 0.0592	1.0243 ± 0.1033	−0.2308 ± 0.000
Average		0.2183 ± 0.1076	0.2203 ± 0.1076	0.1832 ± 0.0748	1.3640 ± 0.1936	−0.0339 ± 0.0823

Note: EHL, Erhualian; FJ, Fengjing; MMS, Middle Meishan; M, Mi; SWT, Shawutou; SJ, Sujiang; SS, Sushan; SMS, Small meishan; H, Huai; SB, Sicilian black pig; LW, Large White; DRC, Duroc.

**Table 3 animals-12-01345-t003:** Nei’s genetic identity (above diagonal) and genetic distance (below diagonal) between 12 pig populations.

ID	EHL	FJ	MMS	M	SWT	SJ	SS	SMS	SB	H	LW	DRC
EHL	****	0.8110	0.8508	0.8289	0.7817	0.6502	0.5656	0.8236	0.5570	0.7769	0.5368	0.5171
FJ	0.2095	****	0.8559	0.8586	0.8344	0.6288	0.5438	0.7854	0.5375	0.7482	0.5133	0.4998
MMS	0.1616	0.1556	****	0.9089	0.9045	0.7114	0.6594	0.8843	0.6435	0.6950	0.6427	0.6243
M	0.1876	0.1525	0.0955	****	0.8935	0.8297	0.7548	0.8596	0.7395	0.8119	0.7342	0.7217
SWT	0.2463	0.1811	0.1003	0.1126	****	0.7990	0.7298	0.8543	0.7176	0.8058	0.7153	0.7167
SJ	0.4305	0.4640	0.3406	0.1867	0.2243	****	0.9538	0.8384	0.9586	0.7704	0.9606	0.9718
SS	0.5699	0.6091	0.4165	0.2813	0.3150	0.0473	****	0.7596	0.9962	0.6680	0.9958	0.9720
SMS	0.1941	0.2416	0.1229	0.1513	0.1575	0.1763	0.2750	****	0.7589	0.7110	0.7571	0.7613
SB	0.5852	0.6209	0.4409	0.3018	0.3318	0.0423	0.0038	0.2758	****	0.6515	0.9965	0.9755
H	0.2525	0.2900	0.3639	0.2084	0.2159	0.2608	0.4035	0.3410	0.4285	****	0.6484	0.6674
LW	0.6221	0.6669	0.4421	0.3090	0.3351	0.0402	0.0042	0.2783	0.0035	0.4332	****	0.9858
DRC	0.6596	0.6936	0.4711	0.3261	0.3331	0.0286	0.0284	0.2727	0.0249	0.4043	0.0143	****

Note: EHL, Erhualian; FJ, Fengjing; MMS, Middle Meishan; M, Mi; SWT, Shawutou; SJ, Sujiang; SS, Sushan; SMS, Small meishan; H, Huai; SB, Sicilian black pig; LW, Large White; DRC, Duroc. ” ****” showed diagonal.

## Data Availability

All data needed to evaluate the conclusions in this paper are present either in the main text or the Appendix A.

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
