# Peer review of "Genetic Evaluation and Population Structure of Jiangsu Native Pigs in China Revealed by SINE Insertion Polymorphisms"

_animals, 2022, doi:10.3390/ani12111345_

Round 1
Reviewer 1 Report
This study investigated genetic evaluation and population structure of Jiangsu native pigs in China revealed by SINE insertion polymorphisms. The genetic characterizations of these pig breeds are important information for the development of conservation, breeding selection, and sustainable utilization. All in all, the experiment design was reasonable and logical, the pig breeds and samples were sufficient, and some valuable results were obtained. However, there are some unsuitable in the writing and expression of the article. In particular, in the Introduction, the concept and identification methods of SINE-RIPs should be added.
Title: In this article, there were not only Jiangsu native pigs in China, but also European pig breeds (Duroc and Large White pigs) and an Italian pig breed (Sicilian Black pigs). Could the authors modify the title to make it more consistent with the content?
Line 18, a period should be before “Therefore”
line 66, there is no need to emphasize "prolific", the article is not about the reproductive performance of Jiangsu native pigs.
Line 96, “native breed (Sicilian black pig) [27] were” Change the semicolon to parentheses. How were the samples of the Italian pig breed obtained?
Line 111-113, How to identify these 18 SINE-RIPs should be explained in detail. This is the key, the most important information for this study. It was not sufficient to provide primers only, as it is necessary to prove that the amplified fragments are short interspersed nuclear elements-short interspersed nuclear elements. For the readers of Animals to understand, it is also necessary to introduce the relevant background knowledge in the Introduction.
Line 120-121, How to determine these sine-RIPS polymorphisms by electrophoresis bands?
Abbreviations for pig breeds on the Figures and Tables should be indicated independently.
Reviewer 2 Report
Manuscript ID #animals-1681307 titled “Genetic evaluation and population structure of Jiangsu native pigs in China revealed by SINE insertion polymorphisms”, by WANG et al.
The authors analysed a set of seven native Chinese pig breeds together with two cosmopolitan pig breeds, two Chinese X Cosmopolitan crossbred pig breeds, and a local Italian pig breed using 18 SINE-RIPs as informative markers.
SINE-RIPs can be considered similar in nature to low polymorphic microsatellites and, therefore, the main results obtained (correct separation between Chinese local and the other pig breeds analysed) do not depart from expectations. However, since the examples illustrating the use of SINE-RIPs for population genetics purposes are scant, I consider that the work presented can be of interest to the readers of Animals.
The work is well written and my concerns can be easily solved by revision:
- The authors stated that the most likely number of K ascertained using STRUCTURE was identified using Structure Harvester. However, this was not presented in the Results section.
I assume that the results from Structure Harvester were flat due to the low polymorphism of the dataset used. However, it would be interesting to reduce the number of K presented in the Results sections (and discussed in the Discussion section to 3). Otherwise, the paragraphs related become verbose and difficult to follow.
- Please include the meaning of the abbreviations of the pig breeds analysed in the Figure captions and, as footnotes in the Tables, when necessary.
Reviewer 3 Report
The present study and its approach appear interesting, though in my opinion it might be of lower impact.
I only have some minor revisions:
- Line 11/24: Why did you use only 18 out of the 30 SINE-RIPs from the previous study? Especially as they are only biallelic, a higher number would have provided more power for the analyses. What criteria were involved in selecting these 18 SINE-RIPs from the previously reported 30.
- I guess you had good reasons not to additionally use (multi-allelic and thus more informative) microsatellites in addition to SINE-RIPs. Please provide your intention in the text.
- Line 18: remove the “a” from utilizations.
- Line 68: This sentence is a little bit unclear and I would suggest rephrasing it, as one third of the mammalian genome originally derived from retrotransposons but they do not account for one third of the genome right now.
- Line 112: The reference should be given as a number.
- Figure 2 – “M” : There seems to be a small size difference between the 8th and 9th lower band – might this be a hint on a third allele? Or can it be explained another way?
- Figures and tables in general: It would be much easier to follow, if you would provide the breed names for each shortcut in a table or figure footnote/legend.
- Table 3: The Note below the table should better be included in the table description above.
- Sujiang and Sushan: How are these crossbreeds mated? Were the external breeds bred to the Chinese breeds at some time in the past and they were just selected afterwards or do matings with the external breeds still occur regularly? In what direction were these crossbreeds selected – towards the external breed? Please include this information in the manuscript.
- Discussion: You state that the population size of Small Meishan pigs has been increased to improve reproductive performance. What is the background of this?
- Discussion: I guess the “using” after “gradually formed…” should be deleted.
